# Antiparasitic Tannin-Rich Plants from the South of Europe for Grazing Livestock: A Review

**DOI:** 10.3390/ani13020201

**Published:** 2023-01-05

**Authors:** Pablo Rodríguez-Hernández, Carolina Reyes-Palomo, Santos Sanz-Fernández, Pablo José Rufino-Moya, Rafael Zafra, Francisco Javier Martínez-Moreno, Vicente Rodríguez-Estévez, Cipriano Díaz-Gaona

**Affiliations:** 1Department of Animal Production, Cátedra de Producción Ecológica Ecovalia-Clemente Mata, UIC ENZOEM, Faculty of Veterinary Medicine, International Agrifood Campus of Excellence (ceiA3), University of Córdoba, Campus de Rabanales, 14071 Córdoba, Spain; 2Animal Health Department (Parasitology and Parasitic Diseases), UIC ENZOEM, Faculty of Veterinary Medicine, International Agrifood Campus of Excellence (ceiA3), University of Córdoba, Campus de Rabanales, 14071 Córdoba, Spain

**Keywords:** phytotherapy, anthelmintics, gastrointestinal nematodes, condensed tannins, organic farming, one health

## Abstract

**Simple Summary:**

The systematic use of different antiparasitic products over time has led to several problems, such as drug resistances and biodiversity loss, which directly affect extensive livestock farming. Considering the current scenario, alternative approaches for parasite control are needed. This review study presents different Mediterranean tannin-rich plants with anthelmintic effect, which can be used as fodder or voluntarily grazed by livestock as a strategy to reduce the use of antiparasitic drugs.

**Abstract:**

Internal parasites are one of the main causes of health threats in livestock production, especially in extensive livestock farming. Despite the environmental toxic effects (loss of dung beetles, biodiversity, and other issues) and resistance phenomenon derived from their prolonged use, anti-parasitic chemical pharmaceuticals are frequently used, even in organic farming. Such a situation within the context of climate change requires urgent exploration of alternative compounds to solve these problems and apparent conflicts between organic farming objectives regarding the environment, public health, and animal health. This review is focused on some plants (*Artemisia* spp., *Cichorium intybus* L., *Ericaceae* family, *Hedysarum coronarium* L., *Lotus* spp., *Onobrychis viciifolia* Scop.) that are well known for their antiparasitic effect, are voluntarily grazed and ingested, and can be spontaneously found or cultivated in southern Europe and other regions with a Mediterranean climate. The differences found between effectiveness, parasite species affected, *in vitro/in vivo* experiments, and active compounds are explored. A total of 87 papers where antiparasitic activity of those plants have been studied are included in this review; 75% studied the effect on ruminant parasites, where gastrointestinal nematodes were the parasite group most studied (70%), and these included natural (31%) and experimental (37%) infections.

## 1. Introduction

Parasites constitute a serious problem for and a major threat to animal production, health, and welfare in both intensive and extensive livestock systems throughout the world, and parasite infections strongly affect livestock, especially in grazing ruminants [1,2]. These infections have an impact on the immune system [3], resulting in lower efficiency due to the reduction of voluntary intake, lower productivity indexes (live weight gain, milk yield, wool production, etc.) [4], lower quality products as a result of considerable protein losses [5], and protein synthesis redirection away from skeletal muscles [6]. All these effects eventually affect animal welfare and farm profitability, leading to serious economic losses and animal deaths [7,8]. The economic burden of major parasitic helminth infections on the ruminant livestock industry in Europe has been estimated at €1.8 billion [9].

In recent decades, parasitic disease control has been mainly achieved by chemotherapy and chemoprophylaxis, through the strategic and repeated administration of antiparasitic drugs [10,11], even in organic farming [12], where these veterinary drugs are allowed [13]. In the particular case of gastrointestinal nematodes (GIN), there exists a worldwide phenomenon of resistance to anthelmintics (AH) due to the exclusive use and reliance on a small range of commercial molecules [14,15,16,17,18]. The rapid development of resistance in GIN populations to synthetic molecules [19] can be seen in the example of monepantel (a relatively new class of AH included between amino acetyl derivates) [20], with cases of resistance described after a few years of commercialisation [21,22]. Furthermore, there has been an increasing number of reports on multidrug resistance to the most commonly used anthelmintic families benzimidazoles, macrocyclic lactones, including moxidectin in sheep, cattle and horses, as well as imidazothiazoles in sheep and tetrahydropyrimidine pyrantel in horses, which all cause concern [23].

In addition to the problem of resistance, the parasite control approach has become a source of public concern in terms of the use of chemical treatments in farm production [24], the potential risk of residues in food such as milk and meat products, and environmental contamination [25,26,27]. A special mention must be made of antiparasitic drugs and their impact on soil and dung fauna biodiversity [28], which also implies an increase in greenhouse gas emissions from grazing livestock faeces [29]. The European project LIFE LiveAdapt pointed out the use of alternative antiparasitic products as one of the best practices towards climate change adaptation for extensive livestock farming [30,31].

All these issues have increased societal demands to reduce the use of chemical compounds in agriculture and livestock breeding [18]. Options for research in alternative approaches to AH drugs include the use of bioactive plants [32], especially in organic farming and sustainable agriculture systems [25,33]. Some *in vivo* and *in vitro* studies have shown that bioactive plants containing different types of secondary metabolites, such as condensed tannins (CT), sesquiterpene lactones, and flavonol glycosides, are a promising option for use in integrated nematode control in farm production systems [34,35,36,37,38,39]. In Mediterranean rangelands, there are many plant species containing secondary metabolites antiparasitic activities, such as sainfoin, sulla, *Lotus* spp., *Cichorium intybus* L., and the *Ericaceae* family rich in CT, and *Artermisia* spp. rich in santonin [39,40,41,42,43,44]. 

Several studies have described the grazing and browsing behaviour of small ruminants, suggesting prophylactic self-medication behaviour in areas with heterogeneous vegetation [45,46]. This strategy is focused on the intake of natural vegetation with its secondary compounds, seeking a lower dependence on conventional chemotherapy in livestock farming [47,48]. In contrast to synthetic anthelmintic drugs, forages or nutraceutical fodders are not imposed but offered to the animals, and can also be used as additives within unifeed rations.

Hence, this review is focused on some plant species, wild or cultivated, and present in European regions with Mediterranean climate which have high content in CT and have been shown to provide some antiparasitic effects in livestock (mainly in ruminants): *Artemisia* spp., *Cichorium intybus* L., *Ericaceae* family, *Hedysarum coronarium* L., *Lotus* spp., and *Onobrychis viciifolia* Scop.). The criteria to select these species are the following: that they have scientific evidence of their antiparasitic effects from experimental procedures, both *in vitro* and *in vivo* with promising potential to be used for parasite control in farms; they are voluntarily grazed and ingested; they are common or widespread; or they have seeds of different cultivars in the market to be cultivated as fodder.

## 2. Plants Containing Secondary Metabolites (PSM) in the Mediterranean Region

PSMs are compounds associated with secondary metabolic pathways, which are non-essential for plant development [49]. Most PSMs serve defence functions, their biosynthesis being linked to biotic or abiotic stimuli [50,51].

This review is mainly focused on CT due to the evidence of antiparasitic activity of tannins-rich forages both *in vivo* and *in vitro* experiences. There are other PSMs implicated, although less information is available on such compounds. Flavanol monomers, flavonol glycosides [52,53], and other phenolic compounds [54] might be involved. Molan et al. [55] showed that flavan-3-ols and flavan-3-ol gallates, the basic units of CT polymers, present an inhibitory activity on egg hatching, larval development, and the viability of *Trichostrongylus colubriformis* third-stage larvae (L3). For example, in sainfoin (*Onobrychis viciifolia* Scop.), a possible action has been attributed to three flavonol glycosides: rutin, narcissin, and nicotiflorin [52]. This is consistent with studies suggesting that the CT present in plant extracts were only partially responsible for the inhibitory effects [56]. 

### 2.1. Condensed Tannins (CT)

In the tannins group, there exist hydrolyzable tannins, such as galic acid [57], and CT, which are polyphenolic compounds contained in various parts of the plant belonging to the flavonoid groups [58]. These molecules have the ability to reversibly bind proteins and other macromolecules [2]. 

Desrues et al. [59] investigated which structural features of CT determine the anti-parasitic effects against the main cattle nematodes (*Ostertagia ostertagi* and *Cooperia oncophora*). They found that the mean degree of polymerization of CT (i.e., average size) and the percentage of PDs within CT seemed to be the most important parameters that influence anti-parasitic activity. Therefore, not only the CT concentration, but also the CT structure influence anthelmintic properties against different life cycle stages of the most important GIN species.

#### 2.1.1. Factors Involved in the Antiparasitic Effect of Condensed Tannins

Evidence by different authors has accumulated over recent decades, suggesting that some bioactive tannin-rich plants have AH effects [44,60]. In addition, other types of parasites, such as coccidia, have been evaluated and it has also been found that the consumption of tanniferous plants, such as lentisk foliage (*Pistacia lentiscus* L.), sainfoin (*Onobrychis viciifolia* Scop.), and sericea lespedeza or sericea (*Lespedeza cuneata* Dum.Cours.) is associated with a reduced number of oocysts per gramme of faeces [61,62,63,64].

In several experiments, the consumption of tanniferous forages was associated with reduced levels of GIN parasites and improved animal performance [65], even when the tannin-rich forages are given as hays or silages [37]. Many studies focused on the AH effects of tannin-rich plants and forages in small ruminants have been published in recent years [1,18,54,66,67,68]. Among these studies, there are interesting findings: a decrease in the establishment of third-stage nematode larvae (L3) or a reduction in worm fertility and egg output in naturally infected goats consuming a moderate concentration of tannin-rich plants [47,69], and a significant and prolonged reduction in nematode egg excretion in naturally infected indoor goats due to a short-term distribution of tannin-rich plants hay [70]. 

It has been hypothesised that the effect of CTs against GIN might be indirect; improvement of the host performance being due to increased protein availability or, directly, by short-term affection of several biological key processes of parasites [71]. This direct influence has been highlighted by many authors both *in vivo* and *in vitro*: CT-containing extracts that could inhibit key parasite enzymes which lead to inhibiting larval development [55,72], larval exsheathment [73,74,75,76,77], larval motility and migration [52,78], egg hatching [55,72], larval feeding [77], and motility of adults [79]. However, the modes of action of CT compounds are currently being studied, and few of them have been elucidated. 

*In vitro* assays have the main advantages of providing easy performance, high reproducibility, rapid results, and low costs. They may also target different stages of the parasite life cycle [80]. In addition, purified compounds isolated from plants can be tested without interference from other plant components or nutrients [81,82].

It is important to emphasise that the results of *in vitro* conditions are not directly applicable to *in vivo* conditions [83]. The gastrointestinal tract activity may substantially modify the active compound of plants, reducing its desirable activity. For example, ruminal conditions could affect tannins because of bacteria degradation [84] and/or the formation of complexes with proteins [54]. Notwithstanding the above, inconsistent results are sometimes obtained from *in vivo* assays, which is why it is important that *in vitro* results are always later evaluated *in vivo* before making general conclusions about antiparasitic properties [83,85].

In the study carried out by Bahuaud et al. [73], the results suggest that the delay or inhibition of the exsheathment of infective larvae (L3) of trichostrongyle nematodes may be related either to the fixation of tannins on proteins of the sheath, disturbing the action of enzymes, or to the complexation with the enzymes themselves. This constitutes an interesting finding, considering that exsheathment is a crucial step in the life cycle, since it represents the transition from the free-living to the parasitic stages [86]. In the case of the experiment *in vitro* conducted by Hansen et al. [87], it was demonstrated that CT inhibited glutathione-S-transferase activity in GIN. Moreover, a direct influence of CTs against parasites is supported by the use of inhibiting substances in many studies. Kabasa et al. [88] observed that the addition of polyethylene glycol (PEG) to the diet of browsing goats, which effectively binds to tannins and inactivates them, led to significant increases in egg excretion and negative effects on growth, suggesting that tannins were involved in the antiparasitic effect. Molan et al. [56] reported the capacity of PEG to partially reverse the antiparasitic effects of a sulla (*Hedysarum coronarium* L.) extract and suggested that CT interfered with the larvae, affecting neurophysiology or neuromuscular coordination. Additionally, AH effects of CT extracts disappear in the presence of polyvinyl polypyrrolidone, a tannin inhibitor, further supporting the conclusion that CTs are AH-active compounds [44,74,75]. 

#### 2.1.2. Factors Affecting the Condensed Tannins Content

Although the grazing of pastures seeded with tanniferous plants has been proposed as a natural way to reduce nematode parasitism, there are also several drawbacks to consider. The effects of CT depend on their concentrations [2] and are influenced by different factors, such as: the type of forage species; the stage of plant maturity [89]; the part of the plant (i.e., leaves vs. stems) [90,91,92]; the leaf:stem ratio, which varies with age and phenological stage [89,93] because most CTs are found in leaves, flowers, and fruits [94]; forage preservation [95]; the season, because environmental factors such as high temperatures, water stress, and extreme light intensities also affect TC concentrations [96]; cultivar and other agronomical factors which alter plant growth and production [2,10]; and even the region [97]. In addition, these plants have differences in the fractions of extractable CT (ECT), protein-bound CT (PBCT), and fibre-bound CT (FBCT), which influence the bioactivity of the CT from legume forage [98]. CTs have been alleged to be responsible for decreases in feed intake, growth rate, feed efficiency, nutrient digestibility [99,100,101], and it might be difficult to ensure that the tanninferous plants contain the optimal levels of CT during peaks of parasitic infections [10]. Furthermore, the efficacy of CT should be confirmed through *in vivo* experiments, not only under controlled laboratory conditions [102]. Therefore, there are many important factors to evaluate before practicing this innovative approach.

#### 2.1.3. Other Effects of Condensed Tannins

Traditionally, CT have been considered anti-nutrient factors because, at high doses, they may have negative effects on feed intake, protein and dry matter digestibilities, live weight gains, milk yield, and wool growth [94,103]. Nevertheless, in recent years, when consumed in moderate concentrations, CT have been recognised as useful phytochemicals for modulating improving growth, milk yields, fertility, and tolerance to some intestinal parasites and arise from the protection of dietary protein from excessive fermentation in the rumen, reducing proteolysis [103,104]. The latter results in a lower concentration of ammonia in the ruminal fluid [105].

However, the CT effect will be considered favourable or detrimental depending on whether or not the positive AH action of these PSMs outweighs their negative nutritional cost to the host [106]. The review by Nawab et al. [107] regarding tannin supplementation in animal feeding shows mitigation strategies by which to overcome the toxic effects of tannins on animal health. Other benefits include bloat prevention [108], which is associated with tannins reducing the stability of a foam that traps ruminal fermentation gases, the effects of tannins on enrichment of conjugated linoleic acids in meat and milk from ruminants, and the inhibition of methanogenesis [89,109]. Moreover, diets with tannins are considered a best practice by which to reduce methane emissions from ruminants [110,111,112].

Therefore, the possible antinutritional consequences of tannin consumption should be simultaneously considered with their positive and antiparasitic effects [113], as well as confirmation in farm-based trials showing reliable economic benefits [102]. The review by Nawab et al. [107] regarding tannin supplementation in animal feeding shows mitigation strategies by which to overcome the toxic effects of tannins on animal health. 

#### 2.1.4. Other Factors Affecting Tannin Action

Along with the evaluation of the tannin’s mechanism, many authors have highlighted the influence of different factors and variables in the CT antiparasitic effect. 

The parasite and host species and the parasite stage are some of the most evaluated [56,79]. For example, when established adult nematodes were exposed to CT in sheep, a reduction in worm fecundity and worm numbers was observed for the intestinal species *Nematodirus battus* and *T. colubriformis*, whereas no changes were recorded for the species that reside in the abomasum (*Teladorsagia circumcincta* and *Haemonchus contortus*) [113,114,115]. However, in goats, the effects on adult worms were restricted to reductions in worm fecundity of *H. contortus* and *T. colubriformis* [10,116], but not of *T. circumcincta* [116]. By contrast, the effects described by Hoskin et al. [117] in red deer suggest that *Trichostrongylus axei* is more susceptible to CT than *Teladorsagia* type nematodes present in the abomasum. This inconclusive and sometimes contradictory background makes it important to test local plants or forages with the most pathogenic or prevalent parasites species of each area to apply this knowledge to practical conditions [118].

The bioactivity of CT *in vivo* against GIN of different livestock species, such as small ruminants, is known to be highly variable depending on the plant source [26], another important factor. The concentration and the structure of the CT present in the different plant species seem to be two major factors modulating efficacy against nematodes. An overview of *in vivo* studies in sheep, goats, and deer involving plants containing CT suggests that a threshold of at least 30–40 g of CT per kg dry matter (DM) (3–4% DM) must be reached to observe antiparasitic activity [26]. Min and Hart [64] also suggested that the observed reduction in faecal egg count (FEC) was generally related to the CT content of forages. 

There have also been studies focused on the structure of monomers of CTs. Molan et al. [55,119] demonstrated that the structure of CT monomers is an important factor in the modulation of their inhibitory effect on the development and the motility of *T. colubriformis* larvae. A similar conclusion was proposed by Brunet and Hoste [53], a hypothesis that is also supported by results from *in vivo* studies which suggested that the tanniniferous plants containing a high prodelphinidin/procyanidin (PD/PC) ratio were most active against gastrointestinal nematodes in ruminants [55,56,65]. In particular, sainfoin is considered to present a high PD/PC ratio [52,120,121].

The influence of forage preservation processes, such as ensiling or pelleting (high temperature and high-pressure treatment), remains unclear. While an alteration of the binding to proteins has been suggested for pelleting [36], it has also been noted that it may also positively influence AH activity [2]. However, in a trial carried out by Gaudin et al. [18] with sainfoin pellets, the efficacy obtained against parasites confirms that the pelleting process of sainfoin does not affect its AH properties, despite possible changes in the binding to proteins. These findings are consistent with results obtained in studies focused on other plant species, such as *Sericea Lespedeza* [36,63,122]. Continuing with sainfoin, different studies have suggested that its AH effect is also maintained in hay and silage [10,69,123]. The preservation of sainfoin, particularly in the form of silage, seems very promising because it can be used for the control of GIN, regardless of the season; more of the plant leaves, which are particularly rich in CTs, are retained than during the drying and conditioning involved in hay-making [123,124].

## 3. Antiparasitic Mechanism of Plants Rich in Secondary Compounds

Although the mechanisms involved in the response of the parasites to the diets containing plants with PSMs are yet unresolved and are currently being investigated, it seems that the bioactive properties of these plants directly arise from their content of PSMs, reducing the level of parasitism and improving the performance of different animal species, such as domestic ruminants [125]. Two potential hypotheses have been put forward to account for the reduction of the adverse effects of parasitism in ruminants grazing, such as forages; while the first hypothesis suggests that PSMs may have direct AH effects on larval and/or adult parasites, interfering in its development through altering the environment in which the nematode is grown [126], the second proposes that the consumption of certain PSMs have host-mediated effects that influence animal biology and improve the immune response to reduce parasitic infestation [126,127]. An example of this mechanism is that the increase in protein availability to the host through protein supplementation via forage during the course of a parasitic infestation can lead to a reduction in the number of nematodes in sheep due to the improvement of their immunity to parasites [104,128,129]. 

However, the response to the consumption of bioactive forages on parasite infections shows great variability due to, among other factors, nematode species, development stages, concentration of PSMs in the plant, prevailing conditions in the digestive tract, and the presence of additional active compounds [54,60,130]. In most of the studies that have compared the value of forages for parasitised animals, it becomes difficult to determine whether the improved physiological response (resilience) is due to improved nutrition or to the role of specific biochemical compounds [26]. However, there is more information supporting the role of some plant metabolites in observed antiparasitic effects [82,131]; unfortunately, most of the studies that have investigated potential antiparasitic effects do not contain sufficient information on the chemical characterisation of the suspected active compounds, making it difficult to accurately attribute activity to specific compounds [26]. 

The beneficial effects of PSMs on host physiology and performance under parasitic challenge have generally been related to the consumption of such bioactive plants when compared with control herbage, such as ryegrass (*Lolium perenne*), white clover (*Trifolium repens*), or lucerne (*Medicago sativa*) [132,133,134]. This effect has been evaluated by: (i) comparison of the clinical status of animals; (ii) pathophysiological measurements; and (iii) assessment of the impact of parasitism on the production of many species, such as sheep, goats, or deer [135,136].

Supplementation with bioactive plants can enhance animals’ ability to regulate the biology of parasite worm populations (resistance), as well as their ability to withstand the negative pathophysiological effects of nematode infections (resilience) [8,137]. The positive effect of these compounds on animal resilience has been underlined in different animal species [36,105,138,139]. 

## 4. Anti-Parasite Plants in Mediterranean Region

The following plant species, wild or cultivated, are present in European regions with a Mediterranean climate, have high content in CT and have an antiparasitic effect on livestock (mainly in ruminants), and are common or widespread or there are commercial seeds of different cultivars in the market to be cultivated as fodder.

The nutritional value is summarised for the species cultivated.

### 4.1. Arthemisia spp.

The genus *Artemisia* belongs to the family *Asteraceae* and is found around the world as a wild plant, but some experiences with “domesticated” and cultivated plants have been carried out [140] and have shown a broad spectrum of medicinal activity, including antiparasitic action [141]. In an Ethnobotany study, Agelet [142] showed *A. absinthium* L. infusion as anthelmintic for livestock. 

The active compounds of *Artemisia* have been characterised [140,141], and some of them have been used in some experiments to demonstrate the activity of each one [131,143]. Artemisinin and its derivates (artemether) are the main compounds studied in antiparasitic effect trials [131,143]. Very high concentrations of pro-anthocyanidins have been found in *Artemisia afra* (1990 mg/100 g) [92] and in *A. herba alba* (2100 mg/100 g) [91]; however, 340 mg/100 g have been found in *A. annua* leaves and only 30 mg/100 g in stems [90]. A paper from Turkey showed 420 mg/100 g of CT in *A. absinthium* [144]. A study from Algeria compared the total extractable tannin content in some browse plant species and found 5700 mg/100 g for *A. campestris* and 3600 mg/100 g for *A. herba alba* [145]. CT in *A. absinthium* significantly varies from one region in Tunisia to another, from 40 to 150 mg/g [97]. 

No effect on livestock performance has been reported for *Artemisia* spp. 

The genus *Artemisia* has an antiparasitic effect not only against nematodes but also against *Fasciola*, and some protozoa are also affected [131,140,141]. The description of trials for antiparasitic activity of *Artemisia* spp. are summarised in Table 1. The main way to provide *Artemisia* to animals is via oral application; however, in fighting against *Fasciola hepatica*, oral application seems not to be effective, while a single intramuscular dose application of extracts of artemether in sheep reduced the egg burden by a 64.9% and worm burden by 91.3% [131].

In the case of protozoa, such as *Leishmania* spp. and *Trypanosoma cruzi*, only *in vitro* experiments have been performed [140,141,146]. Gonzalez-Coloma et al. [140] tried to establish the antiparasitic effect of pure compounds, but they found that the pure compounds had no antiparasitic effect and only extracts from these species had any effect, which may be because of some synergistic components. 

Another parasite which is affected by *Artemisia* consumption is *Trichinella spiralis*, which affects free range pigs. One study reported its activity against the enteral and parenteral phases of *T. spiralis* [147].

**Table 1 animals-13-00201-t001:** Brief description of trials for the antiparasitic effect of *Artemisia* spp.

*In Vivo*/*In Vitro*	Animal Species	Indoor/Outdoor	Natural Infection/Experimental Infection	Parasites	Specific Parasite	Dosage	References
*In vitro*	-	-	-	Protozoa	*Leishmania infantum*, *Trypanosoma cruzi*	800, 400, 100 μg/mL	[140]
					*L. infantum*, *Trypanosoma cruzi*	800, 400, and 100 μg/mL	[141]
					*Leishmania aethiopica* and *Leishmania donovani*	0.0097–0.1565 μL/mL and EC50 0.24–42.00 nl/mL	[146]
	Goat	-	-	GIN	*Haemonchus contortus*	0.5, 1, 2, and 4 mg/mL	[81]
	Sheep	-	-	GIN	*H. contortus*	0.34% DM	[126]
						25 mg/mL and 50 mg/mL	[148]
						CAE 25 mg/mL and CME 25 mg/mL	[149]
						CAE 25 mg/mL and CME 25 mg/mL	[150]
*In vivo*	-	-	Natural infection	GIN	*-*	50 mL/P.O. of A. campestris macerate as single dose	[151]
	Cattle	Outdoor	Natural infection	GIN	*-*	100–150 mg/kg per subject twice in two weeks	[11]
	Gerbil	-	-	GIN	*H. contortus*	160 mg/mL	[143]
	Goat	Outdoor	Natural infection	GIN	*H. contortus* and *Teladorsagia circumcincta*	4 mg/kg of n-hexane extract of A. cina (Acn-h)	[81]
	Mice	Indoor	-	GIN	*Hymenolepis nana*, *Aspiculuris tetraptera*, *Syphacia obvelata*	150 mg/kg	[152]
	Rat	-	-	GIN	*Trichinella spiralis*	300 and 600 mg/kg	[147]
	Sheep	-	-	GIN	*H. contortus*, *Trichostrongylus colubriformis*, *Trichostrongylus axei*, *Oesophagostomum columbianum*, *Strongyloides papillosus* and *Trichuris ovis*	CAE and CME 2.0 and 3.0 g	[149]
					*H. contortus*, *T. ovis*, *Chabertia ovina*, *Bunostomum trigonocephalum y O. columbianum*	CAE and CEE at 1.0 and 2.0 g kg^−1^ BW	[150]
				Platelmint	*Fasciola hepatica*	160 mg/kg (intramuscular dose)	[131]
			Natural infection	GIN	-	4, 6 and 8% DM	[148]
		Indoor	Experimental infection	GIN	*H. contortus*	0.34% DM	[126]

-: Not specified. DM: dry matter EC50: Half maximal effective concentration CAE: crude aqueous extracts CEE: crude ethanolic extracts CME: crude methanol extracts.

### 4.2. Cichorium intybus L.

Chicory (*C. intybus* L.) is a perennial plant of the *Asteraceae* family from the Mediterranean basin, but today it is widely distributed in temperate and semiarid areas around the world. It can be wild or cultivated, and was historically used for multiple purposes, including human consumption [130,153]. There is a review by Li and Kemp [154] about chicory forage yield and animal production. The Puna cultivar of chicory can grow at a rate of >150 kg DM per ha per day in favourable conditions, and herbage production of 7–9 t per ha per year is very common for pure chicory stands for the first two or three years under good grazing conditions [154].

Livestock performance seems to be unaffected by chicory consumption; it can be similar to legume fodder consumption and it can increase yields if it used as a supplement to milk livestock [155], but, at the same time, the compounds (alkaloids, terpenes, saponins, lactones, glycosides, and phenolic compounds) that provoke benefits in animals in higher doses can have antinutritional or detrimental effects [156].

Most of its antiparasitic active compounds have been identified and well-described, e.g., 8-deoxylactucin has been described as a main molecule with antiparasitic activity [130], and other compounds, such as some anthocyanins or flavonoids, have been reported with antiparasitic effects [157].

*In vivo* experiments of chicory have been mainly conducted in sheep, and few in cattle and deer; however, goat *in vivo* studies have not been found for this review. The activity of the chicory produces a reduction in faecal eggs excretion in young animals [33,134,158] and reduces the viability of eggs and larvae stages of different GIN [82,159,160,161]. Although most of the research on the antiparasitic effect of chicory has been conducted against GIN (the description of trials for antiparasitic activity of *C. intybus* are summarised in Table 2), Woolsey et al. [162] found *in vitro* inhibitory effects against *Cryptosporidium parvum*, a protozoon causing production losses and health problems in livestock. This is therefore a promising line of action requiring further research. 

The nutritional value and CT content of *C. intybus* are summarised in Table 3.

### 4.3. Ericaceae Family

Heather species (*Erica* spp., *Calluna vulgaris* (L.) Hull, and others) belong to the *Ericaceae* family and are frequent in the natural vegetation of mountains in humid temperate areas. These species have low nutritive value [94] and a relatively high content of CT (from 30 to 100 g tannic acid equivalents per kg DM) [47]. 

Grazing has been a traditional management method of European heathlands [170]. Heather species have been successfully used in the diets of different livestock species, such as goats, without substantial nutritional cost [105]. Hence, goats seem to be the main species able to include high proportions of heather in their diets [171]. However, López et al. [136] showed that heather-dominated heathlands are unable to maintain profitable grazing systems and to cover the nutritional needs of other species, such as horses. The low nutritive quality of the main plant components of these communities in autumn requires animal access to other more nutritious plant communities or feeding supplementation, in order to enhance foal growth and maintain sustainable horse grazing systems.

The anthelmintic effects of three *Ericaceae* species (*C. vulgaris* (L.) Hull, *Erica cinerea* L., and *Erica umbellata* L.) on different *T. colubriformis* development stages have been shown *in vitro* [54]. 

Furthermore, *in vivo* assays have shown interesting results. Moreno-Gonzalo et al. [172] suggested that heather consumption by goats reduces incoming larvae establishment if it is consumed during the infection of *T. colubriformis*, and reduces FEC by means of a combination between worm burden and female worm fecundity reduction. What is more, studies focused on the supplementation of heather for grazing goats have shown a reduction in the levels of GIN egg excretion associated with a decrease in worm fertility and/or a reduction in the establishment of incoming L3, as well as an apparent greater resilience of goats against gastrointestinal nematode infections [105,132,173,174,175]. However, some limitations associated with these trials should be addressed, such as the lack of heather consumption recorded per animal, the ignorance of total worm burden, and nematode stages involved and/or the strong dependence of weather conditions in the GIN infection dynamics. Additionally, as these studies were conducted in grazing conditions, it was not possible to distinguish whether the reduced worm burden in heather-supplemented goats was due to the reduced degree of pasture contamination (because of the lower levels of egg excretion in faeces) or the direct effect of heather consumption on the mortality and fertility of parasites. The description of trials for antiparasitic activity of *Ericaceae* family is summarised in Table 4.

The inclusion of these plant species, together with areas of improved pastures with tannin-rich grass-legumes, could become an effective tool in achieving the sustainability of animal production systems [176] and animal health in marginal mountain areas.

### 4.4. Hedysarum coronarium L. 

Sulla (*H. coronarium* L.) is a short-lived perennial legume native to the Mediterranean basin where it is grown as a two-year forage crop for grazing and/or hay or silage production [177]. Grazing on sulla forage has been shown to have a positive impact on the productivity of different animal species, such as sheep [178,179,180,181] and goats [182]. These effects have been attributed to its high protein content, degradable-to-structural carbohydrate ratio [183], and moderate content of CT. Its CT content shows large variability, from 8 to 50 g/kg of whole plant DM, depending on the environment, growth stage, and genotype [184]; although these CTs have been pointed out as being responsible for its AH effect, there are studies suggesting that other compounds would also be able to affect the viability of parasites [56]. The description of trials for the antiparasitic activity of the *Ericaceae* family is summarised in Table 5.

This plant is a difficult herbage to agronomically manage because it requires a very specific *Rhizobium* as a nitrogen-fixing bacterial symbiont. In addition, although sulla could be used as a complete diet, it seems feasible to extend the usefulness of a specific pasture by only intermittently grazing it during short periods (at intervals), as it would require some exposure to it, at least on a monthly basis [185]. What is more, it is unlikely that any farm could have a sufficient quantity of this plant for continuous grazing for all animals; hence, pasture plots with a high density of sulla should be managed with rotational grazing.

**Table 5 animals-13-00201-t005:** Brief description of trials for the antiparasitic plant: *Hedysarum coronarium* L.

*In Vivo*/*In Vitro*	Animal Species	Indoor/Outdoor	Natural Infection/Experimental Infection	Parasites	Specific Parasite	Dosage	References
*In vitro*	Deer	-	-	Lung worms	*Dictyocaulus stages viviparus*	1200 μg/mL	[78]
	Sheep	-	-	GIN	*Haemonchus contortus*, *Ostertagia circumcincta* and *Trichostrongylus colubriformis*	50, 100, 200, 400, 800 and 1000 mg CT/mL	[56]
					*T. colubriformis*	200, 400 μg/mL	[72]
					*H. contortus*	3.3% DM	[126]
					*O. circumcincta; T. colubriformis*	3.13, 3.51% DM	[133]
					*T. colubriformis*	400, 800, 1000 μg/mL	[186]
*In vivo*	Deer	Indoor	Experimental infection	GIN	*-*	3.49% DM	[117]
				Lung worms	*Dictyocaulus* sp	3.49% DM	[117]
	Goat	Indoor	Natural infection	GIN	*Teladorsagia circumcincta*, *T.colubriformis*, *T. vitrinus* and *Trichuris* spp.	26 g/kg free CT, 18 g/kg protein-bound CT and 1 g/kg fibre-bound CT	[185]
	Sheep	Indoor	Experimental infection	GIN	*H. contortus*	3.3% DM	[126]
					*O. circumcincta; T. colubriformis*	3.13, 3.51% DM	[133]
					*T. colubriformis*	3% DM	[135]
					*O. circumcincta* and *T. colubriformis*	-	[187]
		Outdoor	Experimental infection	GIN	*T. axei*, *O. circumcincta* and *T. colubriformis (experimento 2)*	9.93 and 12.05% DM	[4]
					*T. colubriformis*	15.8% DM	[125]
					*O. circumcincta; T. colubriformis*	3.13, 3.51% DM	[133]
					*T. circumcincta*	36.9 g/kg DM	[167]
					*T. circumcincta*	15.8 g/kg DM	[168]
					*T. colubriformis* and *O. circumcincta*	-	[188]
			Natural infection	GIN	*T. axei*, *O. circumcincta* and *T. colubriformis (experimento 2)*	9.93 and 12.05% DM	[4]
					*Ostertagia*, *Haemonchus*, *Trichostrongylus*, *Nematodirus*, *Cooperia*	-	[137]
					*T. colubriformis* and *O. circumcincta*	-	[188]
					-	-	[189]

-: Not specified. DM: dry matter CT: condensed tannins.

The nutritional value and CT content of *H. coronarium* is summarised in Table 6. 

### 4.5. Lotus spp.

The genus *Lotus* is formed by perennial legumes adapted to a wide range of weather conditions, including low soil quality and drought [76]. It is usually found as a wild plant, but it can be grown. 

The main active compounds identified in *Lotus* spp. are CT [192,193]. 

The most studied species is birdsfoot trefoil (*L. corniculatus* L.), but other species such as greater bird’s-foot-trefoil (*L. pedunculatus* Cav.) have been included in some experiments [74]. The antiparasitic effect of birdsfoot trefoil has been studied in several experiments, its consumption showing a positive effect in the reduction of FEC or parasite loads when it is consumed fresh (directly grazed or harvested) with a crescent effect depending on the numbers of days eating it [160,193,194]. For example, in the experiment by Cireșan et al. [194], the parasite load was reduced from 1.2% after 7 days to 89.39% after 28 days of consumption. *In vitro* experiments have shown effects on exsheathment inhibition of L3 [76], but it could not be replicated in *in vivo* experiments [83], which implies that more research should be conducted to determine the ways of action and the antiparasitic activity of the *Lotus* genus against GIN. The description of trials for antiparasitic activity of *Lotus* spp. is summarised in Table 7.

The nutritional value and CT content of *L. corniculatus* are summarised in Table 8.

### 4.6. Onobrychis viciifolia Scop.

Sainfoin (*O. viciifolia* Scop.) is a fodder legume forage which can be found in temperate zones, which tolerates drought, cold, and low nutrient status [198]. This fodder is characterised by its appealing features such as high palatability and protein levels [2]. It is widespread in southern Europe and its properties make it very popular on Middle Eastern plateaus and some areas of Spain, Italy, and other countries of Eastern Europe [199]. It has been the subject of renewed interest because it exhibits good nutritional (Table 9) and agronomic qualities beyond its AH properties in such climate conditions [37,123,199,200,201]. Furthermore, sainfoins’ ability to withstand grazing [202] makes it an ideal candidate to provide a therapeutic or preventive antiparasitic treatment.

Its antiparasitic effects have been ascribed to the nature of its secondary metabolites, especially its CT [37,60,123,201]. However, although CT have been its most studied molecules, sainfoin produces other phenolic compounds, such as hydroxybenzoic acids, hydroxycinnamic acids, dihydroflavonols, flavones, flavonols, and flavonol glycosides [203]. 

Sainfoin has shown positive effects against various parasitic nematode species in field studies [70,204]. In general, positive antiparasitic effects have been achieved in intestinal worms, where sainfoin consumption resulted in lower FEC, total egg output [10,70,138,205], and inhibitory activity against L3 of GIN, as measured by larval migration inhibition assay [79,186]. In the case of abomasal parasites, there have been contradictory results; although Paolini et al. [69] found no effect of the intake of sainfoin in goats, Heckendorn et al. [123,200] showed an effect of sainfoin on the same parasite in sheep, when animals were offered sainfoin from 27–28 days post-infection. Sainfoin consumption has been associated with a decrease in nematode fertility or counts in sheep [123,126] and goats [138]. However, the results obtained by Collas et al. [206] in horses do not support any effect of this species on strongyle worm counts or FEC. According to Paolini et al., [138] a repeated distribution of sainfoin hay to grazing dairy goats might be a valuable alternative to reduce nematode infections.

It has been shown that sainfoin extracts not only have an effect against different GIN species in a dose-dependent manner [44,53,201], but also that there are variable CT contents and composition among different sainfoin accessions [207]. This is consistent with different studies which highlighted a lower action of sainfoin against parasites when CT levels were lower than 2% of DM intake [44,123,125,138,208]. However, in a study where different tanniferous plants (*Cichorium intybus* L., *Lotus corniculatus* L.) were tested as antiparasitic against GIN in lambs, sainfoin had the highest content of CT, allowed a higher daily weight gain than the control diet, and, such as the other fodder included in the study, showed antiparasitic activity [60]. The description of trials for antiparasitic activity of *O. viciifolia* is summarised in Table 10.

Future sainfoin cultivation programmes are necessary to evaluate its CT content and composition in order to exploit its full potential as AH forage legume [201].

## 5. Future Research and Conclusions

Overall, the information collected in the present review suggests that future studies must be focused on the investigation of individual parasite species responses towards individual bioactive compounds of different plants in order to learn more about their specific direct action [26]. Moreover, further studies carried out under controlled conditions are needed to determine these changes more precisely. Therefore, it is essential that the first investigations into the effects of an antiparasitic plant be carried out against the most important parasites. Furthermore, the first investigations should be carried out on indicator parasites that serve as laboratory models.

In contrast to synthetic anthelmintic drugs, forages are not imposed but browsed, and can be offered to animals or even used as tannin additives in unified rations to ensure the necessary dose intake. 

Different factors and variables such as the estimation of the daily voluntary pasture intake, the optimal dosages for antiparasitic effect, the days of treatment or intake, and a clarification of the exact action mechanisms must be clarified in order to explain the effects of PSM on livestock nutrition and performance. What is more, the anthelmintic effects of CT-containing fodder used as nutraceuticals depend first on voluntary feed intake, which should be studied to design anthelmintic grazing strategies. 

Torres-Fajardo et al. [215] suggested that the identification of the nutraceutical value of plant species would be a key factor in order to consider developing feeding strategies and management schemes for the sustainable use of pastures. This alternative approach indicates that different natural or cultivated herbages could be used within farm grazing rotations or as fresh or conservated fodders to reduce anthelmintic usage.

## Figures and Tables

**Table 2 animals-13-00201-t002:** Brief description of trials for the antiparasitic plant: *Cichorium intybus* L.

*In Vivo*/*In Vitro*	Animal Species	Indoor/Outdoor	Natural Infection/Experimental Infection	Parasites	Specific Parasite	Dosage	References
*In vitro*	-	-	-	Coccidia	*Cryptosporidium parvum*	300, 150, 75, 37.5, 18.75 and 9.375 μg/mL	[162]
				GIN	*Ostertagia ostertagi*	1000, 500, 250, 100, 50 and 10 µg/mL	[82]
					*Teladorsagia circumcincta*, *Cooperia oncophora* and *Ascaris suum*	7.8 and 500 μg/mL; 1 mg/mL	[130]
					*Caenorhabditis elegans* and *A. suum*	15.6 to 20,000 μg/mL	[161]
					*C. oncophora*	Egg hatch assay: 2500, 1250, 625, 313 and 156 μg/mL Adult motility inhibition assay: 1000, 500, 250, 125, 60 and 30 μg/mL	[163]
					*A. suum* and *Oesophagostomum dentatum*	-	[164]
					*Haemonchus contortus*	1.67–10.03 mg/mL	[165]
	Sheep	-	-	GIN	*H. contortus*, *Trichostrongylus species*, *Trichostrongylus axei*, *Strongyloides* and *Bunstomum*	12.5, 25 and 50 mg/mL	[148]
*In vivo*	Cattle	Indoor	Experimental infection	GIN	*O. ostertagi* and *C. oncophora*	ad libitum	[166]
		Outdoor	Experimental infection	GIN	*O. ostertagi*	ad libitum	[166]
	Deer	Outdoor	Natural infection	GIN	-	0.17 and 0.26% DM	[134]
				Lung worms	-	0.17 and 0.26% DM	[134]
	Sheep	-	Experimental infection	GIN	*Ostertagia*, *Trichostrongylus*, *Oesophagostomu*, *Cooperia*, and *Nematodirus* spp.	2 or 4 kg DM/head per day (green)	[158]
			Natural infection	GIN	*H. contortus*, *Trichostrongylus species*, *T. axei*, *Strongyloides* and *Bunstomum*	12.5, 25 and 50 mg/mL	[148]
		Indoor	Experimental infection	GIN	*H. contortus* and *Cooperia curticei*	3.4 g/kg DM	[60]
		Outdoor	Experimental infection	GIN	*Trichostrongylus colubriformis*	1.49 kg DM/day	[125]
					*T. circumcincta*	8.3 g/kg DM	[167]
					*T. circumcincta*	total phenolic: 262 g/kg DM	[168]
			Natural infection	GIN	*T. circumcincta*, *Trichostrongylus vitrinus*, *T. axei*, *C. oncophora* and *Nematodirus battus.*	total phenolic: 18 and 27 g/kg DM	[33]
					*T. circumcincta*	-	[71]
					*T. circumcita*, *H. contortus*, *C. curticei*, *Trichostrongylus* spp., *Chabertia ovina*, *Oesophagostomum* spp.	-	[159]
					-	-	[160,169]

-: Not specified. DM: dry matter.

**Table 3 animals-13-00201-t003:** Nutritional value of *Cichorium intybus* L. [147].

	Unit	Average
Crude protein	%DM	24.3
Crude fibre	%DM	16.9
Lignin	%DM	0.33
Ash	%DM	18.8
Condensed tannins	%DM	0.17
Sesquiterpene lactones	%DM	0.36

**Table 4 animals-13-00201-t004:** Brief description of trials for the antiparasitic plant: *Ericaceae*.

*In Vivo*/*In Vitro*	Animal Species	Indoor/Outdoor	Natural Infection/Experimental Infection	Parasites	Specific Parasite	Dosage	References
*In vitro*	Goat	-	-	GIN	*Teladorsagia circumcincta* and *Haemonchus contortus*	EC50: 450 μg/mL	[54]
					*H. contortus* and *Trichostrongylus colubriformis*	19% DM	[73]
					*T. colubriformis*	EC50: 120.9, 335.7, 521.6 and 791.3 μg/mL	[118]
*In vivo*	Goat	Indoor	Experimental infection	GIN	*T. circumcincta*	48.2 g tannin acid equivalent/kg DM	[68]
					*T. colubriformis*	64 g tannin acid equivalent/kg DM	[172]
		Outdoor	Natural infection	GIN	*Trichostrongylus*, *Teladorsagia* and *Oesophagostomum*	64 g/kg DM	[105]
					*T. circumcincta*, *Trichostrongylus* spp., and *Chabertia ovina.*	7–8.6% DM	[132]
					*Trichostrongylus* spp., *T. circumcincta*, *Oesophagostomum columbianum*, *Chabertia ovina* and *H. contortus*	84 g tannin acid equivalent/kg DM	[173]
					*Teladorsagia*, *Trichostrongylus* and *Oesophagostomum genera*	30.2–47.2 g tannin acid equivalent/kg DM	[174]
					*T. circumcincta*, *Teladorsagia trifurcata*, *Trichostrongylus axei*, *Trichostrongylus* spp., *C. ovina*, *O. columbianum*, *Trichuris ovis*	61–97 g tannin acid equivalent/kg DM	[175]
	Horse	Outdoor	Natural infection	GIN	-	-	[136]

-: Not specified. DM:dry matter EC50: Half maximal effective.

**Table 6 animals-13-00201-t006:** Nutritional value of *Hedysarum coronarium* L. [190].

	Unit	Average	SD
Dry matter	% as fed	12.3	2.5
Crude protein	%DM	20.2	3.1
Crude fibre	%DM	24.3	4.1
NDF	%DM	36.8	5.8
ADF	%DM	28.8	5.4
Lignin	%DM	8.5	2.0
Ether extract	%DM	2.5	0.4
Ash	%DM	11.4	1.7
Starch	%DM	2.4	
Condensed tannins [191]	%DM	11.97	0.43

**Table 7 animals-13-00201-t007:** Brief description of trials for the antiparasitic plant: *Lotus* spp.

*In Vivo*/*In Vitro*	Animal Species	Indoor/Outdoor	Natural Infection/Experimental Infection	Parasites	Specific Parasite	Dosage	References
*In vitro*	-	-	-	GIN	*Ostertagia ostertagi or Cooperia oncophora*	600, 1200, 1400 μg/mL	[74]
					*Haemonchus contortus*	EC50: 0.66–9.36 mg/mL EC90: 1.48–97.15 mg/mL	[76]
					*H. contortus*	50 mg/mL	[195]
	Sheep	-	-	GIN	*Trichostrongylus colubriformis*	1.6 and 5.5% DM	[135]
*In vivo*	Deer	Indoor	Experimental infection	GIN	-	1.9% DM	[117]
				Lung worms	*Dictyocaulus* sp	1.9% DM	[117]
	Sheep	Indoor	Experimental infection	GIN	*H. contortus* and *Cooperia curticei*	15.2 g/kg DM	[60]
					*H. contortus*	1.8–1.93 kg DM/animal day	[83]
					*T. colubriformis*	1.6 and 5.5% DM	[135]
					*Ostertagia circumcincta* and *T. colubriformis*	-	[187]
					*O. circumcincta* and *T. colubriformis*	ad libitum	[196]
			Natural infection	GIN	-	ad libitum	[194]
		Outdoor	Experimental infection	GIN	*T. colubriformis*	15.9 g/kg DM	[125]
					*Teladorsagia circumcincta*	16 g/kg DM	[168]
					*H. contortus*	13.3–17.4% DM	[197]
			Natural infection	GIN	*Ostertagia*, *Haemonchus*, *Trichostrongylus*, *Nematodirus*, *Cooperia*	-	[137]
					*T. circumcita*, *H. contortus*, *C. curticei*, *Trichostrongylus* spp., *Chabertia ovina*, *Oesophagostomum* spp.	-	[159]
					-	-	[160]
					-	25 g/kg DM	[193]

-: Not specified. DM: dry matter EC50: Half maximal effective concentration EC90: 90% effective concentration.

**Table 8 animals-13-00201-t008:** Nutritional value of *Lotus corniculatus* L. [190].

	Unit	Average	SD
Dry matter	% as fed	23.1	6.8
Crude protein	%DM	21.1	4.2
Crude fibre	%DM	26.4	
NDF	%DM	38.3	8.14
ADF	%DM	28.2	5.5
Lignin	%DM	9.9	3.5
Ether extract	%DM	4.1	0.8
Ash	%DM	9.6	1.9
Condensed tannins	%DM	4.96	2.7

**Table 9 animals-13-00201-t009:** Nutritional value of *Onobrychis viciifolia* Scop [190].

	Unit	Average	SD
Dry matter	% as fed	22.3	3.6
Crude protein	%DM	16.9	2.7
Crude fibre	%DM	25.8	4.9
NDF	%DM	35.4	5.7
ADF	%DM	30.1	4.0
Lignin	%DM	9.4	1.3
Ether extract	%DM	4.1	0.2
Ash	%DM	8.0	1.2
Condensed tannins	%DM	3.0	

**Table 10 animals-13-00201-t010:** Brief description of trials for the antiparasitic plant: *Onobrychis viciifolia* Scop.

*In Vivo*/*In Vitro*	Animal Species	Indoor/Outdoor	Natural Infection/Experimental Infection	Parasites	Specific Parasite	Dosage	References
*In vitro*	-	-	-	GIN	*Ostertagia ostertagi or Cooperia oncophora*	600, 1200, 1400 μg/mL	[74]
	Cattle	-	-	GIN	*C. oncophora* and *O. ostertagi*	10, 40 μg/mL	[201]
	Deer	-	-	GIN		1200 μg/mL	[78]
				Lung worms	*Dictyocaulus stages viviparus*	1200 μg/mL	[78]
	Goat	-	-	GIN	*Haemonchus contortus (mainly)* and *Teladorsagia circumcincta*	1.2 lg/mL	[37,52,209]
					*H. contortus* and *Trichostrongylus colubriformis*	1200 μg/mL	[77]
					*T. circumcincta*, *H. contortus* and *T. colubriformis*	1200 μg/mL	[79]
	Horse	-	-	GIN	*-*	-	[206]
	Sheep	-	-	Coccidia	*Eimeria crandallis*	150, 300, 1200 μg/mL	[61]
				GIN	*T. colubriformis*	200, 400 μg/mL	[72,186]
					*T. circumcincta*, *H. contortus* and *T. colubriformis*	1200 μg/mL	[79]
					*H. contortus*	-	[126]
					*T. colubriformis*	400, 800 and 1000 μg/mL	[186]
*In vivo*	Cattle	Indoor	Experimental infection	GIN	*O. ostertagi* and *C. oncophora*	-	[2]
						2.3% DM	[210]
	Goat	Indoor	Experimental infection	GIN	*H. contortus*	3.5% DM	[69]
			Natural infection	GIN	*Trichostrongyle*		[70]
		Outdoor	Natural infection	GIN	*H. contortus*, *T. circumcincta* and *T. colubriformis*	1.5 kg sainfoin hay/day	[138]
	Horse	Indoor	Natural infection	GIN	*-*	-	[206]
	Rabbit	Indoor	Experimental infection	GIN	*T. colubriformis*	1.8 g/day	[211]
	Sheep	-	Experimental infection	Coccidia	*E. crandallis*	150, 300, 1200 μg/mL	[61]
			Natural infection	Coccidia	*E. crandallis*	150, 300, 1200 μg/mL	[61]
		Indoor	Natural infection	Coccidia	*Eimeria* spp.	-	[198]
			Experimental infection	GIN	*H. contortus* and *T. colubriformis*	-	[1]
					*H. contortus*	600, 1200 and 2400 μg/mL	[18]
					*H. contortus* and *Cooperia curticei*	26.1 g/kg DM	[60]
					*H. contortus* and *T. colubriformis*	1200 μg/mL	[77]
					*H. contortus*		[126]
					*H. contortus* and *C. curticei*	0.13% DM	[200]
					*Haemonchus* spp., *Teladorsagia* spp.*, Nema-todirus* spp. and *Trichostrongylus* spp.	59.71 and 106.62 g/animal day	[208]
					*H. contortus*	8.1 and 9.7% DM	[212]
					*T. colubriformis*	1.211 kg/day of sainfoin	[213]
			Natural infection	GIN	-	15.1 g/kg DM	[214]
		Outdoor	Experimental infection	GIN	*T. colubriformis*	14.9 g/kg DM	[125]
			Natural infection	Coccidia	-	15.1 g/kg DM	[214]
				GIN	-	15.1 g/kg DM	[214]

-: Not specified. DM: dry matter.

## Data Availability

Data sharing not applicable.

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
