# Peer review of "Antiparasitic Tannin-Rich Plants from the South of Europe for Grazing Livestock: A Review"

_animals, 2023, doi:10.3390/ani13020201_

Round 1

Reviewer 1 Report

The manuscript reports a study on the   "ANTIPARASITIC TANNIN-RICH PLANTS FROM THE SOUTH OF EUROPE FOR GRAZING LIVESTOCK: A REVIEW".
The topic could be very interesting; however, the manuscript has serious shortcomings in its exposition and scientific content.

In detail:
- considering that the manuscript is a review, the part on the bibliographic research methodology is missing;

- the part on the chemical description of tannins is very general.  

- information on the vegetative state and whether there is a correlation with the tannin content in plants is missing.

- the values of the tannins present in the treated plants and/or tested in the experimental trials are rarely illustrated;

- many citations refer to the general aspects without going into details on the work reported. This is particularly evident when experimental work on animal and/or in vitro trials is presented;

- dosages and specifics effects are not reported, nor are reference values in the different parasite lifecycles, both in the text and in the tables;

- the tables lack references to the specific parasites on which the plant was tested;

- only a generic positive effect is reported in the text without details;

- references to the tables are not given when the plants considered are discussed in the text, and it is not clear why the nutrition table is given without comment in the text;

- the CT in Tab. 6 is not reported;

- the review should deal with livestock. However, it is not clear why references to mice and gerbils are given unless supported by specific comparison information;

- the possible use of these tannins as additives could also be emphasized.

- In general, the exposition is discursive and non-specific;

The manuscript should focus more on the literature regarding experimental evidence.

Reviewer 2 Report

My comments are attached

Reviewer 3 Report

Dear Authors,

congratulations for your work but I ask you to integrate the Review with other papers in order to make it complete.

Reviewer 4 Report

Antiparasitic Tannin-rich Plants From the South of Europe for Grazing Livestock: a Review. ID animals-1999939

The information contained in this review is important in the field of veterinary parasitology. The use of plants rich in secondary metabolites as anthelmintic has been purposed by several researchers over the world and the results are encouraging.

Some comments and suggestions should be attended to before its publication in this journal

Introduction

L40: Please, delete "..."

L278-278. There are studies of some arboreal legumes with anthelmintic effects, and these works have reported the responsible compounds of the anthelmintic effect. The authors maybe could these findings in this review.

Castillo-Mitre et al., 2017. Caffeoyl and coumaroyl derivatives from Acacia cochliacantha exhibit ovicidal activity against Haemonchus contortus. J. Ethnopharmacology. http://dx.doi.org/10.1016/j.jep.2017.04.010

Zarza-Albarrán et el., 2020. Galloyl flavonoids from Acacia farnesiana pods possess potent anthelmintic activity against Haemonchus contortus eggs and infective larvae. J. Ethnopharmacology. http://dx.doi.org/10.1016/j.jep.2017.04.010

von-Son de Fernex et al., 2017. Anthelmintic effect of 2H-chromen-2-one isolated from Gliricidia sepium against Cooperia punctata. Exp. Parasitol. https://doi.org/10.1016/j.exppara.2017.04.013

Olmedo-Juárez et al., 2022. Phenolic Acids and Flavonoids from Pithecellobium dulce (Robx.) Benth Leaves Exhibit Ovicidal Activity against Haemonchus contortus. Plants. https://doi.org/10.3390/plants11192555

 Table 5 , 7 and 9. Please, write the in vitro e in vivo in italics

Round 2

Reviewer 1 Report

The manuscript is improved, and the tables are more descriptive (specific parasite and dosage). The tables could have been even more comprehensive if they had also reported a summary of the effects. In tables where the reference is missing (e.g. DS), "-" or "NR" (not reported) could have been entered.